# Inclusive Education in Primary and Secondary School: Perception of Teacher Training

**DOI:** 10.3390/ijerph192315451

**Published:** 2022-11-22

**Authors:** Natalia Triviño-Amigo, David Manuel Mendoza-Muñoz, Noelia Mayordomo-Pinilla, Sabina Barrios-Fernández, Nicolás Contreras-Barraza, Miseldra Gil-Marín, Dante Castillo, Carmen Galán-Arroyo, Jorge Rojo-Ramos

**Affiliations:** 1Social Impact and Innovation in Health (InHEALTH) Research Group, Faculty of Sport Sciences, University of Extremadura, 10003 Cáceres, Spain; 2Promoting a Healthy Society Research Group (PHeSO), Faculty of Sport Sciences, University of Extremadura, 10003 Cáceres, Spain; 3Facultad de Economía y Negocios, Universidad Andrés Bello, Viña del Mar 2531015, Chile; 4Public Policy Observatory, Universidad Autónoma de Chile, Santiago 7500912, Chile; 5Centro de Estudios e Investigación Enzo Faletto, Universidad de Santiago de Chile, Santiago 9170022, Chile; 6Physical Activity for Education, Performance and Health (PAEPH) Research Group, Faculty of Sports Sciences, University of Extremadura, 10003 Cáceres, Spain

**Keywords:** inclusive education, educational stages, teacher training, special needs, perception

## Abstract

Introduction. Inclusive education is one of the main objectives of the educational system toward achieving equal opportunities among students. To this end, teacher training plays an important role in the different educational stages. Objectives. To analyze the perceived readiness of teachers for inclusive education and to see the differences in primary and secondary education. Methods: A total of 961 active teachers from public schools, 53.3% Primary and 46.7% Secondary Education, were analyzed by means of a questionnaire on Teachers’ perceptions about their preparation for inclusive education and the CEFI-R instrument. Results. There are statistically significant differences between the two stages in the first questionnaire (question 1: *p* = 0.03; question 2: *p* < 0.01 and question 3: *p* < 0.01) and also, in 3 of the four CEFI-R dimensions, with the primary score being higher. Conclusions: This study shows that there is a large percentage of teachers who believe that their initial training is insufficient to deal with student diversity. In addition, most of them state that continuous training has helped them to improve inclusive education and that they would be willing to attend training courses on inclusion, although in secondary school, the predisposition is lower than in high school. On the other hand, teachers of both educational stages show a mostly favorable attitude according to the CEFI-R, being higher in primary than in secondary school. In this sense, the public administration has work to do.

## 1. Introduction

Inclusive education is defined as an education system that ensures that all students, regardless of their abilities, culture and economic level, have the same opportunities and options to actively participate in school activity, minimizing the risk of exclusion and fulfilling the educational objectives [1,2]. Inclusion is a process of addressing the needs of all students by increasing participation in learning, providing changes in strategies and content, making them appropriate for students [3].

Currently, school classrooms are made up of heterogeneous groups with different needs and students with different abilities, in which educational institutions, and especially teachers, must have the capacity to meet them satisfactorily [2]. Contemporary schools that are considered inclusive value all students equally, increase their active participation in the classroom by building an environment in which no one is excluded considering all the diversity present, reducing the barriers that may appear in the learning of all students, not only those with special needs, and recognizing that inclusion in education is a reflection of an equitable society [4,5].

In schools and institutes lies the responsibility to change both the paradigm and policies that are sometimes exclusive, creating safe spaces where a culture where everyone respects each other is developed [5].

Over the past three decades, there has been a significant movement towards inclusive education worldwide [6], where one of the Sustainable Development Goals (SDGs) developed by the United Nations (UN) for the 2030 Agenda and included in the Incheon Declaration is based on the formation of an inclusive, equitable and quality education system, where regular schools have the necessary resources to serve students with disabilities in the best possible conditions [7,8]. The United Nations Educational, Scientific and Cultural Organization (UNESCO), in 1990, through the World Conference on Education For All, initiated the Education For All (EFA) movement to address the learning needs of all children, youth and adults [9]. This conference resulted in the adoption of the World Declaration on Education For All, which promoted education as a fundamental human right to meet basic learning needs and encouraged countries to implement measures to improve the education system [10].

In this regard, the UN and UNESCO have led this initiative for governments, development agencies, civil society, non-governmental organizations and the media to work towards inclusive education. Several declarations have been issued, such as the Incheon Declaration [8], The Dakar Framework for Action [11] and the Salamanca Statement [3], which reflect the urgent need to provide education for all children, youth and adults with special educational needs [3,8,11].

Other texts, such as the Warnock Report and the Index for Exclusion, attach great importance to the training of special education teachers to include children with Special Educational Needs in their classrooms and emphasize the role of the school in achieving quality education for all, including students with disabilities [5,12].

Organic Law 3/2020, amending Organic Law 2/2006 on Education (LOMLOE) [13], is the state educational legislation in force in Spain. This law proposes the adoption of several approaches in order to reinforce the equity and the inclusive capacity of the system, whose main axis is comprehensive education, making effective the right to inclusive education, for those most vulnerable people, as a human right for all people recognized in the Convention on the Rights of Persons with Disabilities [14] ratified by Spain in 2008 [15]. In the Spanish education system, primary education (between the ages of 6 and 12) is compulsory and consists of three cycles of two academic years each. Secondary education consists of a compulsory part (ESO, Educación Secundaria Obligatoria) of four courses between 12 and 16 years of age. In Extremadura, the autonomous community participating in this research, Decree 228/2014 on Attention to Diversity, regulates educational support for students with special and specific needs [16]. In the Spanish education system, the rate of students in the 2020–2021 academic year with special educational needs resulting from a disability was 2.6%, equivalent to 212,807 students with disabilities with special educational needs [17].

Teachers have an important role in the development of this methodology taking into account the diversity of all their students. Teachers’ attitudes towards inclusion play a very important role in the development of inclusion in the classroom [18,19,20]. In both primary and secondary schools, this variable is key to implementing educational strategies for inclusion [20]. 

In the past, institutions separated students with special needs in order to provide them with special training, separating them from the rest of the students and making their inclusion difficult. Progress has been made towards a methodology in which all students are included within the same teaching process, focusing teaching on the individual child’s learning, changing the perception of “diversity” understood as “disability” [21,22]. 

Teacher-to-teacher support groups [23,24] constitute a community in which they expose their experiences with diversity in order to resolve these situations and create an inclusive culture. On the other hand, barriers to inclusion are not only limited to the economic aspect, but can also arise from culture, language or abilities. To solve these difficulties, communities must be created where teachers, family members, students, caregivers and institutions participate in order to enhance the development of all students [5]. 

Therefore, our study aims to analyze teachers’ perceived readiness for inclusive education and to investigate whether there are differences according to educational stage between primary and secondary education, and, on the other hand, to study the differences in the scores obtained in the four dimensions of the CEFI-R according to educational stage.

It is hypothesized that differences will be found between both educational stages in the perceived initial and continuous preparation for inclusive education and their willingness to attend specific courses on inclusive education; and also, that primary education teachers will score higher than secondary education teachers in all dimensions of the CEFI-R questionnaire, i.e., primary education teachers feel more prepared to address and promote inclusive education than secondary education teachers.

## 2. Materials and Methods

### 2.1. Participants

Participants were selected using a non-probability sampling method based on convenience sampling [25]. The sample consisted of 961 active teachers from public schools in Extremadura, Spain (Table 1). The 53.3% (512) worked as teachers in Primary Education and 46.7% (449) worked in Secondary Education. Of the entire sample, 28.5% (274) were men and 71.5% (687) were women. The median years of experience was 16 years (IQR = 16).

### 2.2. Procedure

To access the sample, the directory of the Ministry of Education and Employment of the Regional Government of Extremadura (Spain) was used to collect the e-mail addresses and telephone numbers of public schools in the region that taught primary or secondary education.

An e-mail was sent to all the educational centers informing them of the aim of the study and it was requested that the message be forwarded to all the teachers in the center. The e-mail included informed consent and a URL link to access the questionnaire. Data were collected between September 2020 and July 2021. 

The questionnaire was administered using the Google Forms tool and was composed of sociodemographic questions, three initial dichotomous questions on initial and continuing teacher training and the CEFI-R questionnaire. 

The questionnaire was administered using the Google Forms tool and was composed of sociodemographic questions, three dichotomous questions on initial training and continuous training received in inclusion and, finally, the CEFI-R questionnaire. The time to complete their participation in the study was estimated at 10 min.

It was decided to use an e-questionnaire because it allowed us to store all the responses in the same database, save costs and obtain a higher response rate by avoiding missing data [26,27].

All data were collected anonymously and kept private. The study was performed according to the guidelines of the Declaration of Helsinki and was approved by the Bioethics and Biosafety Committee of the University of Extremadura (protocol code: 186/2021).

### 2.3. Instruments

Sociodemographic data: Data were taken from a sociodemographic survey containing: gender, age, type of position, educational stage, school location and years of teaching experience.

Teachers’ perceptions of their preparation for inclusive education: This consisted of three dichotomous response questions (question 1: “Do you consider that you were adequately prepared through your initial preparation to respond to the diversity of your students’ needs?”) and questions on lifelong learning (question 2: “Has lifelong preparation helped you to respond to the diversity of your students’ needs?” and question 3: “Would you be willing to attend courses on inclusive education?”).

The Evaluation Questionnaire of Teacher Training for Inclusion, CEFI-R [28]. A total of 19 items falls into four dimensions: dimension 1, “conception of diversity”; dimension 2, “methodology”; dimension 3, “supports”; and dimension 4, “communicative participation”. They are evaluated using a scale whose values range from 1 (strongly disagree) to 4 (strongly agree). The authors reported a Cronbach’s alpha value of 0.79, each factor being above the thresholds of 0.70 as good values (Nunnally and Bernstein, 1994).

### 2.4. Statistical Analysis

The statistical analysis used was the Statistical Package for Social Sciences (Version 26, IBM SPSS, Chicago, IL, USA). The Kolmogorov–Smirnov test was used to confirm that the data showed a normal distribution. Since the assumption of normality was not met, nonparametric tests were used. To calculate the reliability of each of the dimensions of this study, Cronbach’s alpha was used. To analyze the differences between the three dichotomous questions according to the educational stage where the teachers worked, the Person’s chi-square test was used. For the differences between each of the dimensions of the CEFI-R according to the educational stage of the teachers, the Mann–Whitney U test was used; and Spearman’s Rho test was used to analyze the association of each of the dimensions with the different age groups. 

## 3. Results

### The Three Dichotomous Questions

Table 2 shows the responses according to educational stage to the three initial dichotomous questions about the perception of initial and continuous training in inclusion. Statistically significant differences were found in all three questions (question 1: *p* = 0.03; question 2: *p* < 0.01 and question 3: *p* < 0.01).

Table 3 shows the scores of the 4 dimensions of the CEFI-R according to educational stage: Conception of Diversity, Methodology, Supports, and Community Participation. With respect to the first three dimensions (Conception of diversity, Methodology, and Supports), statistically significant differences were found, with primary education teachers scoring higher than secondary education teachers. Regarding the last dimension (Community Participation), although primary education teachers scored higher than secondary education teachers, the difference was not statistically significant.

Table 4 shows the relationship between the different dimensions of the CEFI-R and age ranges, where significant correlations were found in the dimensions of Conception of diversity, supports and community participation. In the methodology dimension, the correlation with respect to the age group was not significant.

## 4. Discussion

### 4.1. Main Findings

The aim of this research is to analyze the differences in the perception of their preparation for inclusive education between primary and secondary school teachers and to study the differences in the scores obtained in the four dimensions of the CEFI-R according to the educational stage.

Among the main findings, the results of the three dichotomous questions show significant differences between teachers of both educational stages. Both felt prepared to meet the needs of their students, with secondary school teachers feeling less prepared (77.5%) than primary school teachers (71.5%). The differences in the second question are much greater (primary 84.2% vs. secondary 72%), with both stating that the training has helped them to meet the needs of their students. In the last question, primary teachers (92.8%) are more participative in receiving training in inclusive education courses than secondary teachers (82.4%). Regarding the CEFI-R scores in its four dimensions, in dimension 1 (Conception of diversity), dimension 2 (Methodology) and dimension 3 (Supports), primary education teachers scored higher than secondary education teachers, these differences being statistically significant. For dimension 4 (Community Participation), the scores were also higher for primary school teachers; however, the differences between the two educational stages were not statistically significant.

The literature on the perception of teacher training in inclusion is limited. In line with the results obtained on the perception of initial training, a 2020 publication showed that teachers needed more training in inclusive content to be able to develop their skills and address student diversity in a satisfactory manner [29], and also, that those teachers who had training other than that required were more aware of inclusion in practice than those who did not [30]. The results show that secondary school teachers perceive that they have worse initial and in-service training than primary school teachers, a fact that is confirmed by the findings of a 2020 study, in which secondary school teachers indicated that they did not have the necessary training to serve all students, since these courses focused on curricular adaptation rather than on putting it into practice [31]. In line with the results obtained in this paper, other research on the initial preparation of secondary school teachers showed that their training was insufficient and that they needed more modules and practices on inclusion in their pre-service training [32]. One of the most important factors in determining the attitude of teachers on inclusion issues is this initial training [33], which has been found to be insufficient in numerous articles, and significant differences are found between educational levels, particularly in secondary school teachers. Refs. [30,31] are as the results of our study.

Regarding the disposition to continuous training on inclusion issues, the attitude of secondary school teachers is significantly lower than that of primary school teachers, results similar to those found in the study by Collado-Sanchís et al., where teachers were less predisposed to all inclusion issues [33]. As suggested in the present research, initial and continuous training towards inclusive education could have an influence by presenting a more positive attitude towards it. In other countries such as Finland, in recent years, there is an increase in negative attitudes towards inclusion, exposing arguments against inclusive education such as increased workload and, above all, not having the necessary skills to offer this type of education, i.e., not feeling self-effective or competent [34]). In Japan, teachers’ attitudes were similar to those in the Savolainen et al. study, with teachers having positive feelings towards interacting with people with disabilities but showing themselves to be very concerned about the inclusion of these children in their classrooms and presenting a really negative attitude [35]. Related to this, in the study by Desombre et al. on French teachers’ attitudes towards inclusive education, general education teachers showed a less positive attitude towards inclusive education than special education teachers, with general education teachers claiming a lower sense of efficacy [36].

In relation to the scores obtained in the CEFI-R questionnaire, in dimension (1), “Conception of diversity”, the results, in general, were high (Me = 3.2). These results show similarities with the results of other studies, suggesting a favorable conception of diversity and inclusion by teachers at present and showing a positive attitude towards inclusive education. [37,38,39,40,41]. However, in geographical areas that present cultural and educational differences with respect to Spain, such as Arabian regions, most teachers show a negative attitude towards it, possibly due to the lack of teacher training on inclusion and diversity through well-designed specific courses, poorly prepared school environment, inadequate curriculum and evaluation modules, among other barriers [42,43,44,45]. In contrast, other research in the Arab population shows that primary school teachers have a better perception of their skills with respect to inclusive education [46]. In other countries such as Finland, researchers have found evidence that Finnish teachers have a high perception of their self-efficacy [47]. In relation to these, several authors suggest that teachers who felt greater self-efficacy and showed a positive attitude towards inclusive education, possessed greater teaching experience, skill and knowledge about inclusive education [48,49,50]. 

In the case of the present research, primary education teachers presented a greater conception of diversity than secondary education teachers, and these differences were statistically significant. Several studies support these results, showing that teachers of pre-school and primary education have more positive attitudes than teachers of secondary education [51,52,53]. Another study reported that secondary school teachers had a better perception of their self-efficacy than elementary school teachers, a surprising finding contrary to the results obtained in this article [54]. This may be due to the fact that teachers of higher educational stages give more importance to the minimum curricular objectives and contents, instead of content and transversal competences, giving more importance to the minimum established than to the students [55].

For dimension (2), “Methodology”, the results for both educational stages are similar (Me = 3), showing that future teachers feel competent to design and carry out the relevant curricular adaptations in the teaching of students with special educational needs. In relation to this, several investigations show how teachers employ daily inclusive strategies and practices in their classrooms, organizing and managing the classroom effectively and designing teaching and assessment strategies adapted to students with special needs [53,56].

Dimension (3), “Supports”, showed the lowest values (Me = 2.4), with primary education teachers scoring higher than secondary education teachers, and these differences were statistically significant. These results are in line with previous investigations (Me = 2.2–2.4) in Spanish teachers using the CEFI-R [39,57]. It can be said that teachers are not entirely in favor of joint planning and collaboration with support teachers or specialists. However, in the study by Pegalajar Palomino et al., 81.2% of secondary school teachers in Jaén were in favor of using specialist professionals for training and advice on inclusive education [41]. According to Ballús et al., the inclusion of the support teacher in the regular classroom can generate multiple benefits, facilitating a shared knowledge in the programming and methodology used according to the needs of the group and, in addition, guaranteeing a more personalized attention [58].

Finally, in dimension (4), “Community participation”, no significant differences were found between the two educational stages. However, it was the dimension that presented the highest score (Me = 3.6). Therefore, teachers in Extremadura would be in favor of the participation of the different agents of the educational community (parents, teachers, school board, etc.). Salinas is in favor of this, giving great importance to the involvement of families to work cooperatively with the teacher, providing positive support during the educational process [59].

To conclude with respect to the CEFI-R questionnaire, it should be noted that there is a significant correlation between the CEFI-R dimensions of Conception of Diversity, Supports and Community Involvement and the age ranges of the teachers. These results could be related to the study by Triviño-Amigo et al. where the three dimensions mentioned above and, in this case, the years of teaching experience, showed a significant correlation with each other [48]. However, in the Methodology dimension, the correlation with respect to age range was not significant in the present investigation, in agreement with Rojo-Ramos et al. [38,57,60].

### 4.2. Practical Applications

Teachers have a framework delimited by the curriculum and educational policies and must program and execute their classes within this term, so sometimes practices are complicated. These teachers must be supported by institutions to develop content effectively; for this reason, teacher training must shift towards better developing the active practical participation of all learners, rather than focusing solely on changing the curriculum for diversity accommodations, and inclusive training must be somewhat more individual in nature, even if it is worked and coordinated collectively [29,31,61]. 

Institutional measures, such as teacher training, workshops and practical training for teachers, and the publication of teaching guides, can assist teachers in the process of acquiring the necessary knowledge about developmental disabilities and teaching methods through practical interaction with children with disabilities before they enter inclusive educational settings [62]. You et al. as well as the present research suggest that these experiences can influence a positive perception of inclusive education and thus enhance their self-efficacy, which is key to effective implementation of inclusive education [40]. One of the factors that determine the attitude of teachers in terms of inclusion is their initial training, so a first training with a well-adapted curriculum is essential for the teacher to have the ability to embrace all the differences of their students [63]. 

In summary, the practical applications drawn from this research are the following: the need to implement quality continuous inclusive training in institutions for new teachers and especially for secondary school teachers; highlighting the importance of initial training so that teachers have the resources and capacity to develop quality inclusive education, increasing their self-efficacy and improving their attitude towards inclusion.

### 4.3. Limitations and Future Lines of Action

The limitations of this study include the following: (1) since it is a cross-sectional study, it is not possible to establish cause—effect relationships; (2) the study participants are limited to Extremadura, so it is likely that there are differences with other communities, since these laws are autonomous. Therefore, in future research, it would be interesting to carry out this type of study in other Spanish autonomous communities, analyzing and comparing teachers’ perception of inclusive education in Spain. (3) Differences by sex and experience were not taken into account in the dichotomous questions or in the CEFI-R. As has been seen in several studies [39,48,49,50], these variables may have an impact on the results. In the future, it would be innovative to consider this study with differences by sex and years of experience. (4) The questionnaires were done online, so there are disadvantages in terms of sampling and response rates.

## 5. Conclusions

The results of the three dichotomous questions show that only 25.7% of the teachers surveyed believe that their initial training is sufficient to deal with the diversity of their students. On the other hand, most of the teachers (78.6%) state that continuous training has helped them to improve inclusive education, and most of them (87.9%) are willing to attend training courses on inclusion, although these last two parameters are lower in secondary education. Teachers of both educational stages show a mostly favorable attitude towards the four dimensions established by the CEFI-R, with primary education teachers showing a more positive attitude than secondary education teachers in the different dimensions.

For these reasons, administrations should implement continuous training and improve initial training to enhance teachers’ self-efficacy and perception of their skills in inclusive education, especially in secondary education teachers.

## Figures and Tables

**Table 1 ijerph-19-15451-t001:** The sociodemographic characterization of the sample.

Variable	Categories	N	%
Sex	Men	274	28.5
	Women	687	71.5
Age	Under 30	71	7.4
	Between 30 and 40	268	27.9
	Between 41 and 50	345	35.9
	Over 50	277	28.8
Position	Internim	244	25.4
	Official	717	74.6
Education stage	Primary education	512	53.3
	Secondary education	449	46.7
Province of the school	Cáceres	333	34.7
	Badajoz	628	65.3
**Variable**		**Me**	**IQR**
Years of experience		16	16

N: number; %: percentage; SD: standard deviation; IQR: interquartile range; Me: Median.

**Table 2 ijerph-19-15451-t002:** Distribution of the three dichotomous questions responses according to educational stage.

			Yes	No	*p*
**(Question 1) Do you think that you were properly prepared through your initial preparation to respond to the diversity of your students’ needs?**					
Education stage	Primary education	N (%)	146 (28.5)	366 (71.5)	0.03 *
	Secondary education	N (%)	101 (22.5)	348 (77.5)	
	Total	N (%)	247 (25.7)	714 (74.3)	
**(Question 2) Has ongoing preparation helped you to respond to the diversity of your students’ needs?**					
Education stage	Primary education	N (%)	431 (84.2)	81 (15.8)	<0.01 **
	Secondary education	N (%)	324 (72.2)	125 (27.8)	
	Total	N (%)	755 (78.6)	206 (21.4)	
**(Question 3) Would you be willing to attend courses on inclusive education?**					
Education stage	Primary education	N (%)	475 (92.8)	37 (7.2)	<0.01 **
	Secondary education	N (%)	370 (82.4)	79 (17.6)	
	Total	N (%)	845 (87.9)	116 (12.1)	

Significant *p*-values are shown in bold. P of the Pearson’s Chi-Square test. The correlation is significant at the ** *p* < 0.01; * *p* < 0.05.

**Table 3 ijerph-19-15451-t003:** CEFI-R descriptive analysis and differences of each Dimension, searching for differences between educational stage.

	Total	Education Stage		
Dimensions	Me (IQR)	Primary Education	Secondary Education	*p*
1. Conception of Diversity	3.2 (1)	3.4 (1)	3 (1)	<0.01
2. Methodology	3 (1.2)	3 (1.35)	3 (1.2)	<0.01
3. Supports	2.4 (0.8)	2.4 (1)	2.2 (0.8)	0.01
4. Community Participation	3.6 (1)	3.8 (0.8)	3.6 (1)	0.06

Me = median value; IQR = Interquartile Range. Each score obtained is based on a Likert scale (1–4): 1 being “Strongly Disagree”, 2 “Partially Disagree”, 3 “Partially Agree” and 4 “Strongly Agree”.

**Table 4 ijerph-19-15451-t004:** Correlations between the dimensions and the age group variable.

Dimensions	Age *ρ* (*p*)
(1) Conception of diversity	−0.16 (<0.01 **)
(2) Methodology	−0.03 (0.26)
(3) Supports	−0.09 (<0.01)
(4) Community Participation	−0.17 (<0.01 **)

The correlation is significant at the ** *p* < 0.01. Each score obtained on the dimensions is based on a Likert scale (1–4): 1 being “Strongly Disagree”, 2 “Partially Disagree”, 3 “Partially Agree” and 4 “Strongly Agree”.

## Data Availability

The datasets used during the current study are available from the corresponding author on reasonable request.

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
