# Peer review of "Inclusive Education in Primary and Secondary School: Perception of Teacher Training"

_ijerph, 2022, doi:10.3390/ijerph192315451_

Round 1
Reviewer 1 Report
On page 3, the second part of this sentence is unclear: In the past, institutions separated students with special needs in order to provide 104 them with special training, thus creating groups within classrooms.
The UNCRPD should be mentioned earlier, not only with reference to its ratification in Spain. The Salamanca Statement should be referred to by name in the text of the article, not only as a reference.
There is a good amount of literature about teachers’ attitudes and it’s important that the article includes a brief overview of the outcomes of this research. There is some discussion of this research literature in the results section, but not for all the tables. The discussion of this literature should be placed in a literature review section earlier in the article. The discussion section which follows the results section can then include brief reference to other research, especially where results are in line with or contrary to those obtained in this research.
On page 7, it is not clear why the research results are being compared with those ‘in Arabian regions’. Why this part of the world and not others?
Author Response
Dear Reviewer:
We appreciate your words about our work and all your comments in order to improve our manuscript.
On page 3, the second part of this sentence is unclear: In the past, institutions separated students with special needs in order to provide 104 them with special training, thus creating groups within classrooms.
This phrase refers to the fact that students with special needs were taught separately according to their needs. They were separated from the rest of the students, so that within the classroom two groups were created, limiting the relationship of these students with the rest of the students. Anyway, we modified the manuscript according to your suggestions, changing it to "In the past, institutions separated students with special needs in order to provide them with special training, separating them from the rest of the students and making their inclusion difficult".
The UNCRPD should be mentioned earlier, not only with reference to its ratification in Spain. The Salamanca Statement should be referred to by name in the text of the article, not only as a reference.
We have added in the introduction the aspects that you have suggested to us. In the case of the UNCRPD, we had previously included both quotations at the end of the sentence. Now, following your suggestion, we have quoted the UNCRPD before and then added the quotation of its ratification in Spain. In addition, we have introduced the Salamanca Statement in the text.
There is a good amount of literature about teachers’ attitudes and it’s important that the article includes a brief overview of the outcomes of this research. There is some discussion of this research literature in the results section, but not for all the tables. The discussion of this literature should be placed in a literature review section earlier in the article. The discussion section which follows the results section can then include brief reference to other research, especially where results are in line with or contrary to those obtained in this research.
Thank you very much, you are right! We have incorporated the comment from table 4 and added paragraph in the discussion. We have also added more bibliographic references as indicated by you.
On page 7, it is not clear why the research results are being compared with those ‘in Arabian regions’. Why this part of the world and not others?
We have found it interesting to compare our results with "Arabian regions" because of the cultural and educational differences with respect to Spain. We have justified this comparison of results in the manuscript.
Reviewer 2 Report
It would be positive to increase the degree of internationalization and updating of bibliographic references, specifically related to inclusive teacher training.
In addition, the discussion could be improved by introducing in a comparative way some more ideas on the practical projection of educational inclusion.
1. What is the main issue addressed by the research?
Inclusive education is a topic of great pedagogical and social interest, with a great scientific projection and topicality.
2. Do you consider the topic to be original or relevant in the field? Is it a topic which addresses a specific gap in the field?
The topic is original and is approached with a view of practical interest for teachers, and also for didactic training.
3. What does it add to the topic in comparison with other published materials? What does it bring to the topic in comparison to other published material?
It brings interesting ideas in relation to inclusion in education, taking into account differences in its development at different educational stages.
4. What specific improvements should the authors consider with regard to methodology? What other controls should be considered?
I have not suggested any improvements in Methodology, but I have in the discussion and conclusions section. It is necessary to improve the degree of contrast, comparison and differentiation of ideas and pedagogical positions on educational inclusion with a higher degree of internationalisation (European, world context, etc.).
5. Are the conclusions consistent with the evidence and arguments presented and do they address the main question posed? and do they address the main question posed?
Yes, the conclusions are consistent with the study overall. Perhaps, as a recommendation that could improve the manuscript, it would be good to provide some more 'specific'light or more concrete proposals for improving inclusive teacher education.
6. Are the references adequate?
Yes, but they could be made more international and up to date.
7. Please include any additional comments on tables and figures.
All correct.
Author Response
Dear reviewer:
Thank you very much for your kind words about our manuscript, as well as for all your efforts to contribute to improving the presentation of our research in it.
It would be positive to increase the degree of internationalization and updating of bibliographic references, specifically related to inclusive teacher training.
In addition, the discussion could be improved by introducing in a comparative way some more ideas on the practical projection of educational inclusion.
These issues he suggests will be addressed in the following sections. Let us hope that we have responded correctly to your suggestions.
- What specific improvements should the authors consider with regard to methodology? What other controls should be considered?
I have not suggested any improvements in Methodology, but I have in the discussion and conclusions section. It is necessary to improve the degree of contrast, comparison and differentiation of ideas and pedagogical positions on educational inclusion with a higher degree of internationalisation (European, world context, etc.).
Following your comment, we have included a paragraph in the discussion section contrasting data from different studies and countries. Hopefully we have addressed your suggestion
- Are the conclusions consistent with the evidence and arguments presented and do they address the main question posed? and do they address the main question posed?
Yes, the conclusions are consistent with the study overall. Perhaps, as a recommendation that could improve the manuscript, it would be good to provide some more 'specific’ light or more concrete proposals for improving inclusive teacher education.
According to your indications, we have added a reference supporting these proposals in the practical applications section.
- Are the references adequate?
Yes, but they could be made more international and up to date.
As noted in question 4, we have added more comparative references from international studies.
New references added:
Ahsan, M. T., Sharma, U., & Deppeler, J. M. (2012). Exploring Pre-Service Teachers’ Perceived Teaching-Efficacy, Attitudes and Concerns About Inclusive Education in Bangladesh. International Journal of Whole Schooling, 8(2), 1-20.
Alnahdi, G. H., & Schwab, S. (2021). Special Education Major or Attitudes to Predict Teachers’ Self-Efficacy for Teaching in Inclusive Education. Frontiers in Psychology, 12. https://www.frontiersin.org/articles/10.3389/fpsyg.2021.680909
Coates, J. K. (2012). Teaching inclusively: Are secondary physical education student teachers sufficiently prepared to teach in inclusive environments? Physical Education & Sport Pedagogy, 17(4), 349-365. https://doi.org/10.1080/17408989.2011.582487
Desombre, C., Lamotte, M., & Jury, M. (2018). French teachers’ general attitude toward inclusion: The indirect effect of teacher efficacy: Educational Psychology: Vol 39, No 1. Educational Psychology, 39(1). https://doi.org/doi.org/10.1080/01443410.2018.1472219
Savolainen, H., Engelbrecht, P., Nel, M., & Malinen, O.-P. (2012). Understanding teachers’ attitudes and self-efficacy in inclusive education: Implications for pre-service and in-service teacher education. European Journal of Special Needs Education, 27(1), 51-688.
Savolainen, H., Malinen, O.-P., & Schwab, S. (2022). Teacher efficacy predicts teachers’ attitudes towards inclusion – a longitudinal cross-lagged analysis. International Journal of Inclusive Education, 26(9), 958-972. https://doi.org/10.1080/13603116.2020.1752826
Yada, A., & Savolainen, H. (2017). Japanese in-service teachers’ attitudes toward inclusive education and self-efficacy for inclusive practices. Teaching and Teacher Education, 64, 222-229. https://doi.org/10.1016/j.tate.2017.02.005
Round 2
Reviewer 1 Report
Dear authors
Thanks very much for making the recommended changes.